# A cross-sectional study of the psychological impact of the COVID-19 pandemic on camped refugees in Ghana

Barbara Sakyi[1], Fiifi Amoako Johnson[1,2]*

1 Faculty of Social Sciences, Department of Population and Health, College of Humanities and Legal Studies, University of Cape Coast, Cape Coast, Ghana, 2 Faculty of Social Sciences, Centre for Mixed Migration and Diaspora Studies (CeMMiDS), College of Humanities and Legal Studies, University of Cape Coast, Cape Coast, Ghana

* famoakojohnson@ucc.edu.gh

**Data Availability Statement:** The data relevant to this study are ethically restricted as they contain sensitive and potentially identifying information. The data underlying the results presented in this

## Abstract

Vulnerable populations such as camped refugees are often exposed to spread of infectious diseases because of their living conditions, limited resources available to them and exclusion from social services. This study examined the psychological state of camped refugees in Ghana during the COVID-19 pandemic and how their background characteristics predict the severity of the pandemic's psychological impact. It covered 763 refugees aged 15 years and above resident in two (Krisan = 316 and Ampain = 447) camps. Nine COVID-19 Anxiety Scale indicators were used to examine the psychological state of camped refugees. A composite indicator was derived to examine the overall psychological impact. Logistic regression was used to examine the factors that were associated with severe psychological impact. The multivariate analysis revealed that sex of the respondent, marital status and age of head of household were the only socio-demographic factors associated with having a severe psychological impact of the pandemic. There was very strong evidence that respondents who had moderate (OR = 1.74, 95% CI = 1.12, 2.7) and high (OR = 1.66, 95% CI = 1.05, 2.63) knowledge of the disease had increased odds of severe psychological impact. Also, those with moderate (OR = 2.97, 95% CI = 1.78, 4.97) and high (OR = 12.98, 95% CI = 7.86, 21.42) adherence had increased odds of severe psychological impact. None of the pre-existing health conditions and challenges were not significantly associated with severe psychological impact. The limited number of significant socio-demographic covariates suggests that severe psychological impact of the pandemic was a problem in the general population, and thus interventions should target the general population of camped refugees. Also, health education should not only focus on enhancing knowledge and promoting preventive measures but also on managing psychological distress.

## Introduction

COVID-19 was declared a global pandemic by the World Health Organization (WHO) on the 11th of March 2020 [1,2]. Since then, globally, according to the World Health Organisation

study are available upon reasonable request through the University of Cape Coast Ethical Review Board: Email: irb@ucc.edu.gh, Address: University of Cape Coast Ethical Review Board, University of Cape Coast, PMB, Cape Coast, Ghana.

**Funding:** The authors received no specific funding for this work.

**Competing interests:** The authors have declared that no competing interests exist.

(WHO), there have been 546,357,444 confirmed cases and 6,336,415 deaths as of 5[th] July 2022 [3]. Nonetheless, studies that have analysed excess mortality due to COVID-19 have reported global deaths to be about three times higher than officially reported [4,5]. In Ghana, the first case of COVID-19 was reported on 12[th] March 2020. Since then, the Ghana Health Service has reported 166,546 confirmed cases and 1,452 deaths as of 5[th] July 2022 [6].

The spread of infectious diseases such as COVID-19 has been a global concern, particularly to vulnerable populations such as camped refugees because their living conditions render them vulnerable [7,8]. These include overcrowding, poor living and working conditions, sharing of washing and sanitation facilities, poor standards of hygiene, inadequate access to food and medical provisions and limited access to employment opportunities [7]. In addition to the aforementioned, millions of refugees have been exposed to traumatic and life-threating experiences pre-departure and on their journeys, compounded by challenges of social isolation and integration into their host countries [9]. COVID-19 posed a new threat, which coupled with their already fragile state has the potential to aggravate their psychological wellbeing [10]. Likewise, containment measures implemented during pandemics do not target the needs of refugees, particularly those camped, but rather exacerbates their plight as they are unable to engaging in economic activities due to restrictions on movement and inability to access healthcare and other essential services owing to exclusion from national welfare schemes [11]. Evidence suggests that although these preventive measures are important for slowing the spread of infectious diseases such as COVID-19, they are difficult to adhere to as they have significant social impacts, particularly among vulnerable populations such as camped refugees with limited resources and dependent on humanitarian aid [12–15]. Decline in humanitarian assistance due to limited activity of agencies, the fear and anxiety associated with the disease has the potential to impact the psychological wellbeing of camped refugees [16–21].

Despite the high exposure of camped refugees to infectious diseases and their consequential impacts, their psychological wellbeing is largely overlooked, even in emergencies [22,23]. In spite of these disadvantages and the acknowledged weak health systems in Africa, the United Nations (UN) Sustainable Development Goals (SDG 3) stipulates by 2030, the need to achieve good health and wellbeing for all. Health systems challenges and poor living conditions poses challenges to achieving SDG 3, particular for camped refugees due to the structural vulnerabilities and social isolations they are exposed to.

In Sub-Sahara Africa, studies have reported that civil conflicts, disease epidemics, and poverty severely impact the psychological wellbeing of refugees [24]. However, the health needs of refugees are often overlooked [25], although the continent is host to almost one-third of the world's 29.4 million refugees and asylum-seekers [26]. Similarly, although Ghana host 12,215 refugees' of which 5,659 are resident in five (Buduburam, Ampain, Krisan, Fentana, and Egyeikrom) camps across the country [27,28], the literature shows that they experience challenges including disruption in social networks, disputes over resources, high rate of unemployment, overcrowding, lack of access to proper health care, water shortages, inadequate sanitation facilities, and increased level of criminal activities [29]. These conditions expose camped refugees in the country to acquisition and spread of infectious diseases [7,8,30–32]. The aforementioned, in addition to fear, anxiety and uncertainty during pandemics such as COVID-19 has the potential to impact the psychological wellbeing of camped refugees in the country. In this regard, this study examined the psychological state of camped refugees during the COVID-19 pandemic in Ghana and how socio-demographic characteristics, COVID-19 related factors, pre-existing health conditions and challenges within the camps are associated with severe psychological impact of the pandemic.

The study was conducted in the Ampain and Krisan refugee camps in the Western region of Ghana. These camps were selected because of their differential population and cultural

dynamics. The Krisan is a multi-national camp hosting 808 refugees from 17 African countries and Pakistan, whilst the Ampain camp is the biggest camp in Ghana hosting 3306 refugees from Cote d'Ivoire. Researching the psychological wellbeing of camped refugees has the potential to inform policy and practice on addressing the challenges they face in times of pandemics.

## Methodology

### Study population and design

The study population was camped refugees aged 15 years and above. Given the retrospective nature of the study, the target population was limited to those aged 15 years and above to avoid recall bias, ensure information provided were accurate and reflected real life experiences. The data was collected between 29th July and 6th August 2021 to examine the conditions of camped refugees in Ghana during the COVID-19 pandemic. The data was collected in two refugee camps (Krisan and Ampain) in the Western region of Ghana. Ethical approval for the study was obtained from the University of Cape Coast Institutional Review Board with approval number UCC/IRB/CHLS2021/20 and with permission from the Ghana Refugee Board. Informed consent, explaining the objectives of the study and its relevance and assuring respondents of anonymity and confidentiality, was sort before interviews were conducted. Respondents who could read were made to read the informed consent statement themselves before signing them. For those who could not read, the informed consent was read to them and where necessary interpreted before signing or thumbprinting. Where respondents were below 18 years of age, informed consent was sort from their parents/guardians.

The study considered potential ethical issues that could arise and initiate measures to mitigate their effects. There was the potential risk of respondents being exposed to emotional distress if they have had negative experiences of COVID-19. In such situation, arrangements were made for respondents to be referred to the social welfare counsellor whom the GRB has officially assigned as the camp counsellor. Field assistants were trained on ethical issues relating to seeking informed consent, respect for privacy, anonymity and confidentiality, and how to notice emotional distress among participants, including mood swings, exhaustion, helplessness, and or hopelessness. Given that Research Assistants could also experience physical, psychological, and social risks when undertaking their responsibilities, they also received training in community entry protocols, approaching and interviewing respondents in a respectful manner and how to deal with hostile situations when they arise.

According to data from the Ghana Refugee Board [27], the Krisan camp host 615, while the Ampain camp host 2409 refugees aged 15 years and above. The study aimed to collect data from a representative sample of respondents from both camps. The sample size was derived using the Yamane (1967) formula [33]. A sample size of 243 was derived for the Krisan camp and 344 for the Ampain camp. Considering the special nature of the target population and the potential for high non-response rate, an additional 30% was added to each sample, resulting in a sample size of 316 and 447 for the Krisan and Ampain camps, respectively.

A two-stage sampling approach was used to select respondents. At the first stage of sampling, an estimate of the number of houses in each camp was derived through a reconnaissance exercise. The estimated number of housing unit were 474 and 686 in the Krisan and Ampain camps, respectively. A systematic sampling approach was used to select housing units. For each camp, a sampling fraction of the number of housing unit to the sample size was computed–Krisan Camp = 474/316 = 1.5, approximated to 2, and Ampain Camp = 686/446 = 1.54, also approximated to 2. To select the first house in each camp, a random number between 1 and 2 was generated, and the sampling fraction added to the random number to select subsequent houses. At the second stage of sampling, the selected housing units were visited and

eligible respondents identified. In order not to potentially violate the assumption of statistical independence [34,35], where more than one eligible respondent was identified in a housing unit, the lottery method was used to randomly select one respondent. The instrument for data collection was pre-tested at the Egyeikrom refugee camp in the Central Region of Ghana.

## Outcome

The indicators selected to examine the psychological wellbeing of camped refugees during the COVID-19 pandemic were aimed to understand their emotional, personal, and behavioural state. In this regard, literature on the COVID-19 Anxiety Scale (CAS) were reviewed [36–41]. CAS is an instrument developed to collect information to measure dysfunctional anxiety associated with COVID-19. It is important to note that, there is no one standard CAS tool for examining mental health. Researchers have developed CAS instruments to meet the objectives and context of their research. Based on the review of CAS and in the context of camped refugees, nine CAS questions were developed and used to examine the psychological state of camped refugees during the COVID-19 pandemic in Ghana (Table 1). Each question was rated on a 5-point Likert scale to reflect their severity, ranging from 1 = never, 2 = rarely, 3 = sometimes, 4 = very often to 5 = always. We used the Cronbach's alpha to assess reliability of the data. The nine indicators yielded a Cronbach's alpha of 0.907 indicating high reliability.

A composite indicator was derived using factor analysis to examine the overall psychological state of camped refugees during the COVID-19 pandemic. The technique involves a mathematical procedure that transforms correlated variables into a smaller number of uncorrelated variables called factor scores [42]. The factor scores are a linear combination of the original variables which are derived in decreasing order of importance with the first factor score accounting for as much of the variability in the data as possible, and each succeeding factor accounting for as much of the remaining variability. The higher the factor loading the higher the contribution of the indicator to the factor scores (composite index). To conduct the factor analysis, the indicators (Table 1) were recoded into 1 if the respondent indicated that they had the effect of interest very often or always and 0 otherwise. The first factor loadings (Table 1) were all positive, indicating that the first factor score represented overall psychological effect of the pandemic on camped refugees. As about 30.0% or more of respondents indicated having

**Table 1. COVID-19 Anxiety Scale questions for measuring psychological impact of COVID-19.**

| Question | First factor loadings |
|---|---|
| 1. At the peak of the coronavirus pandemic in Ghana, did you feel dizzy, lightheaded, or faint, when you read or listened to news about the disease? | 0.571 |
| 2. Did you have trouble falling asleep during the peak of the COVID-19 outbreak in Ghana? | 0.729 |
| 3. How often did you feel so hopeless that you did not want to carry on living during the peak of the COVID-19 outbreak in Ghana? | 0.756 |
| 4. How often were you unable to carry out essential activities for daily living because of the feelings of fear of COVID-19? | 0.789 |
| 5. During the peak of the COVID-19 pandemic, how often were you distressed, disturbed or upset because you were inactive and unable to carry out essential daily activities? | 0.762 |
| 6. How often did reminders of the virus caused you to have physical reactions, such as sweating or a pounding heart during the peak of COVID-19? | 0.795 |
| 7. To what extent were you worried about coming into contact with other refugees for fear of being infected? | 0.528 |
| 8. To what extent were you worried of contracting the disease because of the living arrangements in the camp? | 0.616 |
| 9. To what extent were you worried of contracting the disease because of stigmatisation? | 0.605 |

the event of interest (Table 1) either very often or always, to examine the characteristics of the most severely affected, the first factor scores (composite index) was ranked and the top 30.0% coded as "1" to represent camped refugees who were severely impacted psychologically by the COVID-19 pandemic and "0" otherwise.

## Covariates

The covariates selected for the analysis was based on the literature [7,8,30–32,43–45]. The covariates were categorised into four groups–socio-demographic characteristics, COVID-19 related factors, pre-existing health conditions and challenges within the camps. The socio-demographic characteristics included camp of residence, country of origin, sex, age, marital status, educational attainment, religious affiliation, engagement in economic activity, head of household, sex and age of the household head. The COVID-19 related factors were the number of sources of information on COVID-19, preferred source of information on COVID-19, level of knowledge on COVID-19 and adherence to the preventive measures. Factor analysis was used to derive an index of knowledge on COVID-19 and also adherence to COVID-19 using the variables shown in Tables 2 and 3, respectively. The first factor loadings for both indices were all positive, indicating that they represented overall level of knowledge and also adherence to the COVID-19 preventive measures (Tables 2 and 3). The composite indices were categorised into tertiles to reflect camped refugees with low, moderate and high knowledge of COVID19 and also low, moderate, and high adherence to the COVID-19 preventive measures. The covariates also covered pre-existing health conditions (diabetes, hypertension, heart disease and chronic lung disease) of respondents and challenges faced within the camps (overcrowding, water shortages, lack of soap, and loss of livelihood).

## Statistical analysis

Cross-tabulation was used to explore the bivariate relationship between the selected covariates and severe psychological impact of the COVID-19 pandemic. The analysis was conducted with

**Table 2. Indicators used to measure knowledge of COVID-19.**

| Indicator | First factor loading |
|---|---|
| 1. COVID-19 is highly contagious | 0.177 |
| 2. Transmission /Contaminated respiratory droplets | 0.374 |
| 3. Transmission /Contaminated surfaces/objects | 0.344 |
| 4. Not all people who contract COVID-19 show symptoms | 0.321 |
| 5. Symptoms show within 2–14 days of infection | 0.262 |
| 6. Children aged between 0 and 5 years can be infected | 0.373 |
| 7. Young people aged between 10 and 24 years can be infected | 0.274 |
| 8. Infected person can spread the virus even if they show no symptoms | 0.259 |
| 9. Symptoms include cold | 0.351 |
| 10. Symptoms include stuffy nose | 0.281 |
| 11. Symptoms include sneezing | 0.386 |
| 12. Symptoms include feverishness | 0.492 |
| 13. Symptoms include headaches | 0.564 |
| 14. Symptoms include body pains | 0.583 |
| 15. Symptoms include loss of smell | 0.639 |
| 16. Symptoms include loss of taste | 0.596 |

**Table 3. Indicators used to measure adherence to the COVID-19 preventive measures.**

| Indicator | First factor loadings |
|---|---|
| 1. Practised social distancing | 0.747 |
| 2. Limited contact with other people | 0.782 |
| 3. Wore nose/face mask while outside the camp | 0.726 |
| 4. Wore nose/face mask while with visitors | 0.745 |
| 5. Wore nose/face mask when in public places | 0.708 |
| 6. Washed hands with soap under running water for at least 20 seconds after touching surfaces or objects | 0.811 |
| 7. Washed hands with soap under running water for at least 20 seconds after shaking hands or avoided shaking hands | 0.863 |
| 8. Washed hands with soap under running water for at least 20 seconds before eating | 0.347 |
| 9. Washed hands with soap under running water for at least 20 seconds after visiting public places or avoided public places | 0.739 |
| 10. Used sanitizers to disinfect hands after touching surfaces or objects | 0.834 |
| 11. Used sanitizers to disinfect hands after shaking hands or avoided shaking hands | 0.824 |
| 12. Used sanitizers to disinfect hands before eating | 0.720 |
| 13. Used sanitizers to disinfect hands before touching mouth, eyes or nose | 0.850 |

the pooled data and independently for each camp. Chi-Squared test was used to examine statistically significant differences. Fisher's Exact test was used where cell sample sizes were small (less than five) [46]. Within and between camp statistical significant differences were also examined. Logistic regression analysis was used to examine the factors that were statistically significantly ($p < 0.05$) associated with severe psychological impact of the COVID-19 pandemic amongst camped refugees in Ghana. A sequential modelling approach was used to examine the socio-demographic characteristics, COVID-19 related factors, pre-existing health conditions and the challenges within the camps that were statistically significantly associated with severe psychological impact of the COVID-19 pandemic on camped refugees in Ghana. Model 1 accounted for the socio-demographic characteristics. Model 2 included the COVID-19 related factors, whilst Models 3 and 4 added the pre-existing health conditions and the challenges, respectively.

## Results

### Psychological state of camped refugees during the COVID-19 pandemic

A high percentage (75.4%, 95% CI = 72.3, 78.4) of camped refugees reported feeling dizzy, lightheaded, or faint, when they read or listened to news on COVID-19 (Fig 1). The results further showed that, almost half of respondents reported having sleeping difficulties (49%, 95% CI = 45.7, 52.8), whilst a similar percentage (45.6%, 95% CI = 42.1, 49.2) were distressed or disturbed and unable to carry out essential daily activities. Further, 31.9% (95% CI = 35.6, 42.5) reported being unable to carry out essential activities for daily living because of fear of COVID-19. Also, 29.1% (95% CI = 25.9, 32.3) felt hopeless and did not want to carry on living during the peak of the COVID-19 pandemic, with a similar percentage reporting physical reaction such as sweating or heart palpitations. With regards to apprehension, 38.0% (95% CI = 34.6, 41.5) were apprehensive coming into contact with other refugees, 31.7% (95% CI = 28.4, 35.0) were worried contracting the disease because of stigma and 30.9% (95% CI = 27.7, 34.2) because of living arrangements in the camps.

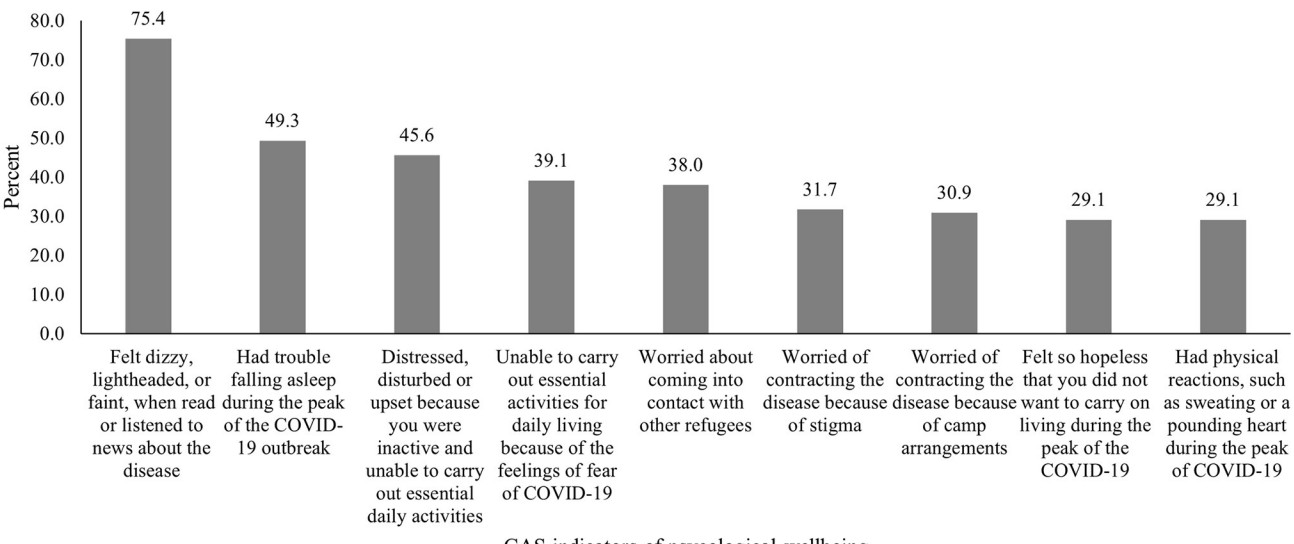

**Fig 1. Percentage distribution of respondents by CAS indicators of psychological wellbeing.**

## Bivariate analysis of the background characteristics of camped refugees and severe psychological impact of the COVID-19 pandemic

Table 4 shows the percentage distribution of respondents who were severely psychologically impacted by the COVID-19 pandemic by their socio-demographic characteristics, COVID-19 related factors, pre-existing health conditions and challenges within the camps. The results of the bivariate analysis showed that severe psychological impact of the COVID-19 pandemic did not vary significantly (p = 0.245) for refugees resident in the Ampain camp (32.2%, 95% CI = 27.8, 36.5) when compared to those in the Krisan camp (28.2%, 95% CI = 23.2, 33.2). Also, the psychological impact of the pandemic was not statistically significantly different with regards to country of origin. Furthermore, analysis of the pooled data and the within and between camp analysis revealed that religious affiliation, engagement in economic activity, and household characteristics such as being head of the household, sex and age of head of household were not significantly associated with severe psychological impact of the pandemic. Severe psychological impact of COVID-19 was also not related to preferred source of information on COVID-19. Similarly, the results from the pooled data showed that being severely affected psychologically by the pandemic was not dependent on having a pre-existing health condition. However, the between camp analysis revealed that a statistically significantly higher percentage of refugees with pre-existing health conditions in the Ampain camp were severely psychologically impacted when compared to those in the Krisan camp. Considering the challenges, only loss of livelihood was statistically significantly (p = 0.009) associated with having a severe psychological distress.

The socio-demographic characteristics observed to be statistically significantly associated with being severely impacted psychologically by the COVID-19 pandemic were the sex, age, educational attainment, and marital status of the respondent. The results showed that a higher percentage of female (33.5%, 95% CI = 29.3, 38.3) camped refugees were significantly more likely to have had severe psychological impact when compared to their male (26.3%, 95% CI = 21.5, 31.0) counterparts. The within camp analysis revealed that sex differentials of the psychological impact of the pandemic were statistically significantly different (p = 0.023) for refugees resident in the Krisan Camp, with a higher percentage of females being more affected

**Table 4. Percentage distribution of respondents who had severe psychological impact of the COVID-19 pandemic by socio-demographic characteristics, COVID-19 related factors, pre-existing health conditions and challenges within the camps.**

| Background characteristics | Pooled data | | | Ampain Camp | | | Krisan Camp | | | Between camp p-value |
|---|---|---|---|---|---|---|---|---|---|---|
| | % [95% CI] | n | p-value | % [95% CI] | n | p-value | % [95% CI] | n | p-value | |
| **Overall** | 32.2 [27.8, 36.5] | 763 | NA | 32.2 [27.8, 36.5] | 451 | NA | 28.2 [23.2, 33.2] | 312 | NA | 0.245 |
| **Socio-demographic characteristics** | | | | | | | | | | |
| Country of origin | | | 0.380 | | | NA | | | 0.850 | |
| Cote d'Ivoire | 32.2 [28.0, 36.4] | 478 | | 32.7 [28.3, 37.0] | 451 | | 26.5 [11.4, 41.5] | 34 | | 0.457 |
| Sudan/South Sudan | 26.3 [18.8, 33.8] | 133 | | — | | | 26.9 [19.3, 34.6] | 130 | | |
| Other | 28.9 [21.7, 36.2] | 152 | | — | | | 29.7 [22.3, 37.1] | 148 | | |
| Sex | | | 0.026 | | | 0.354 | | | 0.023 | |
| Female | 33.8 [29.3, 38.3] | 432 | | 33.8 [28.2, 39.5] | 269 | | 33.7 [26.5, 41] | 163 | | 0.985 |
| Male | 26.3 [21.5, 31.0] | 311 | | 29.7 [23.0, 36.3] | 183 | | 22.1 [15.5, 28.8] | 149 | | 0.122 |
| Age (in years) | | | 0.002 | | | 0.003 | | | 0.412 | |
| 15–19 | 21.1 [15.6, 26.6] | 213 | | 20.6 [13.8, 27.4] | 136 | | 22.1 [12.8, 31.4] | 77 | | 0.798 |
| 20–29 | 32.8 [26.0, 39.6] | 186 | | 35.2 [25.3, 45.0] | 91 | | 30.5 [21.2, 39.8] | 95 | | 0.501 |
| 30–39 | 30.3 [22.9, 37.6] | 152 | | 33.3 [23.5, 43.1] | 90 | | 25.8 [14.8, 36.8] | 62 | | 0.321 |
| 40+ | 38.2 [31.7, 44.8] | 212 | | 41.0 [32.7, 49.4] | 134 | | 33.3 [22.8, 43.9] | 78 | | 0.265 |
| Educational attainment | | | 0.787 | | | 0.098 | | | 0.012 | |
| No formal education | 32.0 [23.0, 41.1] | 103 | | 41.8 [28.7, 55.0] | 55 | | 20.8 [9.2, 32.4] | 48 | | 0.023 |
| Primary | 29.5 [25.0, 33.9] | 404 | | 33.3 [27.5, 39.2] | 249 | | 23.2 [16.6, 29.9] | 155 | | 0.030 |
| Secondary or higher | 31.6 [25.9, 37.3] | 256 | | 26.5 [19.4, 33.7] | 147 | | 38.5 [29.4, 47.7] | 109 | | 0.033 |
| Marital status | | | 0.002 | | | 0.001 | | | 0.301 | |
| Ever married | 36.9 [31.5, 42.3] | 306 | | 42.2 [34.6, 49.9] | 161 | | 31.0 [23.5, 38.6] | 145 | | 0.043 |
| Never married | 26.3 [22.2, 30.3] | 457 | | 26.6 [21.5, 31.6] | 290 | | 25.7 [19.1, 32.4] | 167 | | 0.851 |
| Religious affiliation | | | 0.172 | | | 0.532 | | | 0.197 | |
| Christian | 31.8 [28.0, 35.6] | 585 | | 31.9 [27.5, 36.3] | 433 | | 31.6 [24.2, 39] | 152 | | 0.947 |
| Other | 26.4 [19.9, 32.9] | 178 | | 38.9 [15.7, 62.1] | 18 | | 25 [18.3, 31.7] | 160 | | 0.205 |
| Engaged in economic activity | | | 0.591 | | | 0.602 | | | 0.901 | |
| No | 31.1 [27.2, 35.1] | 530 | | 32.8 [27.8, 37.7] | 351 | | 27.9 [21.3, 34.5] | 179 | | 0.256 |
| Yes | 29.2 [23.3, 35.0] | 233 | | 30.0 [21.0, 39.0] | 100 | | 28.6 [20.9, 36.3] | 133 | | 0.812 |
| Head of household | | | 0.689 | | | 0.422 | | | 0.815 | |
| No | 29.8 [25.1, 34.6] | 362 | | 30.4 [24.4, 36.4] | 227 | | 28.9 [21.2, 36.6] | 135 | | 0.762 |
| Yes | 31.2 [26.6, 35.7] | 401 | | 33.9 [27.7, 40.1] | 224 | | 27.7 [21.1, 34.3] | 177 | | 0.180 |
| Sex of head of household | | | 0.672 | | | 0.762 | | | 0.591 | |
| Female | 29.6 [24.2, 35.0] | 277 | | 31.4 [24.6, 38.1] | 185 | | 26.1 [17.1, 35.1] | 92 | | 0.366 |
| Male | 31.1 [27, 35.2.0] | 486 | | 32.7 [27.1, 38.4] | 266 | | 29.1 [23.1, 35.1] | 220 | | 0.391 |
| Age of head of household | | | 0.057 | | | 0.284 | | | 0.289 | |
| Less than 30 | 31.3 [23.7, 38.8] | 144 | | 34.7 [23.6, 45.8] | 72 | | 27.8 [17.4, 38.2] | 72 | | 0.369 |
| 30–39 | 23.3 [16.8, 29.8] | 162 | | 24.1 [14.6, 33.5] | 79 | | 22.6 [13.6, 31.6] | 84 | | 0.829 |
| 40–49 | 35.9 [29.9, 41.9] | 245 | | 36.0 [28.6, 43.5] | 161 | | 35.7 [25.4, 46.0] | 84 | | 0.962 |
| 50+ | 29.4 [23.2, 35.5] | 211 | | 30.9 [23.2, 38.6] | 139 | | 26.4 [16.1, 36.6] | 72 | | 0.492 |
| **COVID-19 related factors** | | | | | | | | | | |
| Number of sources of information on COVID-19 | | | <0.000 | | | 0.016 | | | 0.001 | |
| One | 30.7 [24.5, 36.9] | 215 | | 35.6 [27.9, 43.3] | 149 | | 19.7 [10.0, 29.4] | 66 | | 0.020 |
| Two | 19.4 [13.7, 25.0] | 186 | | 21.7 [14.3, 29.1] | 120 | | 15.2 [6.4, 23.9] | 66 | | 0.282 |
| Three or more | 36.2 [31.2, 41.1] | 362 | | 36.3 [29.3, 43.3] | 182 | | 36.1 [29.1, 43.1] | 180 | | 0.976 |
| Preferred source of information on COVID-19 | | | 0.677 | | | 0.424 | | | 0.731 | |
| Television | 30.9 [27.2, 34.5] | 622 | | 33.0 [28.2, 37.8] | 370 | | 27.8 [22.2, 33.3] | 252 | | 0.169 |

(*Continued*)

**Table 4.** (Continued)

| Background characteristics | Pooled data | | | Ampain Camp | | | Krisan Camp | | | Between camp p-value |
|---|---|---|---|---|---|---|---|---|---|---|
| | % [95% CI] | n | p-value | % [95% CI] | n | p-value | % [95% CI] | n | p-value | |
| Other | 29.1 [21.6, 36.6] | 141 | | 28.4 [18.5, 38.3] | 81 | | 30.0 [18.3, 41.7] | 60 | | 0.836 |
| Knowledge of COVID-19 | | | 0.003 | | | 0.001 | | | 0.211 | |
| Low | 24.4 [19.1, 29.7] | 254 | | 26.1 [19.3, 32.9] | 161 | | 21.5 [13.1, 29.9] | 93 | | 0.413 |
| Moderate | 38.0 [32.1, 44.0] | 255 | | 44.7 [36.4, 52.9] | 141 | | 29.8 [21.4, 38.3] | 114 | | 0.015 |
| High | 29.1 [23.5, 34.7] | 254 | | 26.8 [19.7, 34.0] | 149 | | 32.4 [23.4, 41.4] | 105 | | 0.339 |
| Adherence to the COVID-19 preventive measures | | | <0.000 | | | <0.000 | | | <0.000 | |
| Low | 9.8 [6.2, 13.5] | 254 | | 8.0 [3.8, 12.2] | 162 | | 13.0 [6.1, 20.0] | 92 | | 0.197 |
| Moderate | 24.1 [18.8, 29.4] | 253 | | 29.6 [21.6, 37.6] | 125 | | 18.8 [12.0, 25.5] | 128 | | 0.004 |
| High | 57.4 [51.4, 63.5] | 256 | | 57.9 [50.3, 65.5] | 164 | | 56.5 [46.3, 66.7] | 92 | | 0.827 |
| **Pre-existing health condition** | | | | | | | | | | |
| Diabetic | | | 0.973 | | | 0.215 | | | 0.029 | |
| No | 30.6 [27.0, 34.1] | 661 | | 31.0 [26.3, 35.6] | 378 | | 30.0 [24.7, 35.4] | 283 | | 0.800 |
| Yes | 30.4 [21.4, 39.4] | 102 | | 38.4 [27.1, 49.6] | 73 | | 10.3 [0.0, 21.6] | 29 | | 0.006 |
| Hypertensive | | | 0.366 | | | 0.107 | | | 0.252 | |
| No | 30.0 [26.6, 33.5] | 689 | | 30.8 [26.2, 35.4] | 393 | | 29.1 [23.9, 34.2] | 296 | | 0.623 |
| Yes | 35.1 [24.2, 46.1] | 74 | | 41.4 [28.6, 54.2] | 58 | | 12.5 [0.0, 29.2] | 16 | | 0.032 |
| Heart disease | | | 0.982 | | | 0.285 | | | 0.031 | |
| No | 30.6 [27.1, 34.0] | 671 | | 31.2 [26.5, 35.8] | 382 | | 29.8 [24.5, 35.0] | 289 | | 0.698 |
| Yes | 30.4 [21.0, 39.9] | 92 | | 37.7 [26.2, 49.2] | 69 | | 8.7 [0.0, 20.5] | 23 | | 0.009 |
| Chronic lung disease | | | 0.791 | | | 0.380 | | | 0.057 | |
| No | 30.7 [27.2, 34.2] | 661 | | 31.3 [26.6, 36.0] | 380 | | 29.9 [24.5, 35.3] | 281 | | 0.695 |
| Yes | 29.4 [20.5, 38.3] | 102 | | 36.6 [25.3, 47.9] | 71 | | 12.9 [0.9, 24.9] | 31 | | 0.016 |
| **Challenges within the camp** | | | | | | | | | | |
| Overcrowding | | | 0.477 | | | 0.088 | | | 0.225 | |
| No | 31.0 [27.5, 34.6] | 651 | | 34.0 [29.1, 38.8] | 365 | | 27.3 [22.1, 32.4] | 286 | | 0.067 |
| Yes | 27.7 [19.4, 36.0] | 112 | | 24.4 [15.3, 33.6] | 86 | | 38.5 [19.4, 57.5] | 26 | | 0.161 |
| Water shortages | | | 0.454 | | | 0.083 | | | 0.163 | |
| No | 31.1 [27.5, 34.6] | 647 | | 34.2 [29.2, 39.2] | 351 | | 27.4 [22.3, 32.5] | 296 | | 0.062 |
| Yes | 27.6 [19.4, 35.8] | 116 | | 25.0 [16.5, 33.5] | 100 | | 43.8 [18.6, 68.9] | 16 | | 0.119 |
| Lack of soap | | | 0.769 | | | 0.916 | | | 0.746 | |
| No | 30.4 [26.9, 33.8] | 671 | | 32.3 [27.5, 37.0] | 372 | | 28.0 [22.9, 33.1] | 300 | | 0.239 |
| Yes | 31.9 [22.2, 41.5] | 91 | | 31.6 [21.3, 42.0] | 79 | | 33.3 [5.5, 61.2] | 12 | | 1.000 |
| Loss of livelihood | | | 0.009 | | | 0.236 | | | 0.018 | |
| No | 31.6 [28.2, 35.0] | 722 | | 32.8 [28.3, 37.2] | 430 | | 29.8 [24.5, 35.0] | 292 | | 0.395 |
| Yes | 12.2 [2.1, 22.3] | 41 | | 19.0 [1.8, 36.3] | 21 | | 5.0 [0.0, 14.8] | 20 | | 0.343 |

when compared to males. However, the differences were not significantly different for those resident in the Ampain camp. The results further shows that sex differentials of the psychological impact of the pandemic were not significantly different between the camps. Regarding age, a higher percentage of older camped refugees (40 years or older) were severely psychologically impacted by the COVID-19 pandemic when compared to those younger (Table 4). The within camp analysis revealed that the age differentials were statistically significantly different (p = 0.003) for refuges in the Ampain camp but not (p = 0.412) for those in the Krisan camps. Considering the between camp effects, the age differentials of the psychological impacts were not significantly between each of the age categories.

The pooled analysis showed that the psychological impact of the COVID-19 pandemic did not vary significant different by educational attainment of the respondents. The within camp analysis revealed that the educational differentials of the psychological impact were not statistically significant for respondents in the Ampain camp. However, they were statistically significant (p = 0.012) for those in the Krisan camp, with a higher percentage of those with secondary or higher education (38.5%, 95% CI = 29.4, 47.7) being psychologically impacted when compared to those with primary (23.2%, 95% CI = 16.6, 29.9) and no formal education (20.8%, 95% CI = 9.2, 32.4). The between camp analysis, showed that a significantly higher percentage of respondent in each age category in the Ampain camp were severely psychologically impacted by the pandemic when compared to those in the Krisan camp (Table 4). Also, camped refugees who had ever married (36.9%, 95% CI = 31.5, 42.3) were statistically significantly (p = 0.002) more likely to be affected psychological when compared to those who had never married (26.3%, 95% CI = 22.2, 30.3). The analysis for the individual camps showed that a statistically significantly (p = 0.001) higher percentage of ever married respondents in the Ampain camp (42.2%, 95% CI = 34.6, 49.9) were severely affected by the pandemic when compared to those never married (26.6%, 95% CI = 21.5, 31.6). With regards to the Krisan camp the differences for those ever married when compared to those never married were not (p = 0.301) statistically significantly different. Further, the between camp analysis showed that a significantly higher percentage of those ever married in the Ampain camp were severely impacted by the pandemic when compared to those in the Krisan camp. With regards to those never married, the differences were statistically significantly different for respondents in the Ampain camp when compared to those in the Krisan camp.

Regarding the COVID-19 related factors, the results showed that a higher percentage of respondents who get information on COVID-19 from three or more sources were severely psychologically impacted by pandemic than those with lesser sources of information. Similar findings were observed for both the Ampain and Krisan camps (Table 4). Considering the between camp analysis, the findings shows that a statistically significantly (p = 0.020) higher percentage of refugees with one source of information on COVID-19 in the Ampain camp (35.6%, 95% CI = 27.9, 43.3) were severely psychologically impacted by the pandemic when compared to those in the Krisan camp (19.7%, 95% CI = 10.0, 29.4).

Further, the results showed a very strong evidence that respondents who were more knowledgeable (p = 0.003) about the disease and adherent (p < 0.001) to the preventive measures were those most impacted psychologically. Considering the level of knowledge, 38.0% (95% CI = 32.1, 44.0) of those with moderate knowledge and 29.1% (95% CI = 23.5, 34.7) of those with high knowledge were severely affected when compared to 24.4% (95% CI = 19.1, 29.7) of those with low knowledge. For the Ampain camp, a statistically significantly (p = 0.001) higher percentage (44.7%, 95% CI = 36.4, 52.9) of those with moderate knowledge were psychologically impacted by the pandemic when compared to those with low (26.1%, 95% CI = 19.3, 32.9) and high (26.8%, 95% CI = 19.7, 34.0) knowledge of COVID-19. Nonetheless, for the Krisan camp, the higher the level of knowledge, the higher the percentage severely psychologically impacted by the pandemic, but the differences were not large enough to be statistically significant (p = 0.211). Between the camps, the results shows that the percentage severely psychologically impacted by the pandemic does not vary significantly for those with low and high knowledge, however, a significantly (p = 0.015) higher percentage of those with moderate knowledge in the Ampain camp (44.7%, 95% CI = 36.4, 52.9) were impacted when compared to those in the Krisan camp (29.8%, 95% CI = 21.4, 38.3). More than half (57.4%, 95% CI = 51.4, 63.5) of respondents who were highly adherent to the preventive measures were severely impacted psychologically by the pandemic, when compared to those who were moderately (24.1%, 95% CI = 18.8, 29.4) and least adherent (9.8%, CI = 6.2, 13.5). Similar findings

were observed for the independent analysis for the Ampain and also the Krisan camp. The cross-camp analysis showed that the percentage severely psychologically impacted by the pandemic did not vary significant for respondent with low and high adherence in the Ampain and Krisan camps, however, a statistically significantly (p = 0.004) higher percentage of those with moderate adherence in the Ampain camp (29.6%, 95% CI = 21.6, 37.6) were psychologically impacted when compared to those in the Krisan camp (18.8, 95% CI = 12.0, 25.5).

The pooled analysis revealed that the psychological impact of the pandemic were not significantly different for those with and without pre-existing health conditions (Table 4). Within camps, the analysis revealed that the psychological impact of the pandemic were not significantly different for those with and without pre-existing health conditions, except for those with diabetes and heart diseases in the Krisan camp where a significantly (p = 0.029 and 0.031, respectively) lower percentage of those with the condition were impacted when compared to those without the condition (Table 4). The between camp analysis further showed that a significantly higher percentage of those with pre-existing health condition in the Ampain camp were severely psychologically impacted by the pandemic when compared to those with a pre-existing health condition in the Krisan camp. The differentials for those without pre-existing health condition in the Ampain and Krisan camps were not large enough to be statistically significantly different (Table 4).

Considering challenges within the camps, the results showed that a statistically significantly (p = 0.009) lower percentage (12.2%, 95% CI = 2.1, 22.3) of respondents who lost their livelihood were severely impacted when compared to those who did not lose (31.6%, 95% CI = 28.2, 35.0) their livelihood. A statistically significantly (p = 0.018) lower percentage (5.0%, 95% CI = 0.0, 14.8) of refugees in the Krisan camp who lost their livelihood were severely impacted when compared to those who did not (29.8%, 95% CI = 24.5, 35.0) lose their livelihood. Although the percentages were higher (lost their livelihood = 32.8%, 95% CI = 28.3, 37.2, did not lose their livelihood = 19.0%, 95% CI = 1.8, 36.3) for the Ampain camp, the differences were not statistically significant (p = 0.236). The between camp differences were not statistically significantly different.

## Factors associated with severe psychological impact of the COVID-19 pandemic

Binary logistic regression models were fitted to examine the factors that were statistically significantly associated with severe psychological impact of the COVID-19 pandemic. A sequential modelling approach was used to examine the background characteristics that were statistically significantly associated with the psychological state of camped refugees during the COVID-19 pandemic in Ghana. Model 1 accounted for the socio-demographic characteristics. Model 2 included the COVID-19 related factors, whilst Models 3 and 4 added the pre-existing health conditions and the challenges, respectively. Table 5 shows the estimated odds ratios of being severely impacted psychologically by the pandemic and the statistically significant background characteristics of the respondents, along with their corresponding 95% confidence intervals.

When the socio-demographic covariates were included in the model (Model 1), the sex of the respondent, marital status, and the age of the head of household were observed to be statistically significantly (p < 0.05) associated with severe psychological impact of the COVID-19 pandemic. However, when the COVID-19 related factors were included in the model (Model 2), the sex of the respondent became statistically insignificant (p > 0.05), but the other socio-demographic covariates (Model 1) remained significant alongside the number of sources of information on COVID-19, level of knowledge on COVID-19 and the level of adherence to

**Table 5. Factors associated with the severe psychological impact of the COVID-19 pandemic.**

| Background characteristics | Model 1 OR [95% CI] | Model 2 OR [95% CI] |
|---|---|---|
| **Socio-demographic characteristics** | | |
| Sex | | |
| Female | 1.00 | |
| Male | 0.71 [0.51, 0.97] * | |
| Marital status | | |
| Ever married | 1.00 | 1.00 |
| Never married | 0.61 [0.44, 0.83] ** | 0.60 [0.42, 0.87] ** |
| Age of head of household | | |
| Less than 30 | 1.00 | 1.00 |
| 30–39 | 0.57 [0.34, 0.95] * | 0.49 [0.27, 0.87] * |
| 40–49 | 1.05 [0.67, 1.65] | 0.87 [0.52, 1.45] |
| 50+ | 0.78 [0.48, 1.25] | 0.65 [0.38, 1.11] |
| **COVID-19 related factors** | | |
| Number of sources of information on COVID-19 | | |
| One | | 1.00 |
| Two | | 0.56 [0.33, 0.94] * |
| Three or more | | 1.25 [0.82, 1.90] |
| Level of knowledge of COVID-19 | | |
| Low | | 1.00 |
| Moderate | | 1.74 [1.12, 2.70] * |
| High | | 1.66 [1.05, 2.63] * |
| Level of adherence to the COVID-19 protoccols | | |
| Low | | 1.00 |
| Moderate | | 2.97 [1.78, 4.97] ** |
| High | | 12.98 [7.86, 21.42]** |
| **Likelihood ratio test** | | |
| Deviance (-2 log likelihood) | 916.478 | 756.223 |
| Change in deviance (Model 1 –Model 2) | | 160.255 |
| Degrees of freedom | | 5 |
| p-value associated with change in deviance | | <0.001 |

the preventive measures. None of the pre-existing health conditions and the challenges were observed to be statistically significant (p > 0.05) when they were included in Models 3 and 4. The likelihood ratio test was used to examine the model (Model 1 versus Model 2) of best fit. When the COVID-19 related factors were included in the model, the deviance (-2 loglikelihood) reduced by 160.3, with an associated p-value less than 0.000. The large decline in deviance and the p-value implies that Model 2 is a better fit with higher predictive power than Model 1. Therefore, interpretation of the results was based on Model 2.

The results presented in Table 5 shows that the socio-demographic factors, marital status and the age of head of household and the COVID-19 related factors, number of sources of information on COVID-19, level of knowledge and adherence to the COVID-19 preventive measures were statistically significantly associated with severe psychological impact of the pandemic.

Table 5, Model 2 shows that respondents who were never married had reduced odds of 0.60 (95% CI = 0.42, 0.87) of having severe psychological impact of the COVID-19 pandemic when compared to their counterparts who had ever married. Regarding the age of the head of

household, the results showed that respondents from households where the head was 30 years or older, were less likely to have severe psychological impact when compared to where the respondent was from a household where the head was less than 30 years. The differences were more pronounced and statistically significant between households where the head was 30–39 years and less than 30 years. In this regard, respondents from households where the head was aged between 30–39 years had reduced odds of 0.48 (95% CI = 0.27, 0.86) of having severe psychological impact when compared to where the head was less than 30 years.

Considering the COVID-19 related factors, the results showed that camped refugees who had information on COVID-19 from two sources were statistically significantly ($p < 0.05$) less likely (OR = 0.58, 95% CI = 0.35, 0.97) to have had severe psychological impact during the pandemic when compared to those who had information from one source or more than three sources. The results further showed very strong evidence that level of knowledge on COVID-19 and adherence to the preventive measures were associated with severe psychological impact of the pandemic. Camped refugees who were knowledgeable of the disease and more adherent to the preventive measures were those severely impacted psychologically by the pandemic. The results revealed that those who were highly and moderately knowledgeable about the disease had increased odds of 1.66 (95% CI = 1.05, 2.63) and 2.97 (95% CI = 1.78, 4.97), respectively, of being severely impacted psychologically when compared to those who had low knowledge of the disease. Similarly, those who were highly adherent to the preventive measures had increased odds of 12.98 (95% CI = 7.82, 21.42) of having a severe psychological impact, whilst those who were moderately adherent had increased odds of 3.10 (95% CI = 1.85, 5.17), when compared to those who had low adherence.

## Discussion

The study investigated the psychological impact of the COVID-19 pandemic on camped refugees in Ghana and the background characteristics that were statistically significantly associated with severe psychologically impact of the pandemic. Examining the psychological impact of the pandemic on vulnerable populations such as camped refugees was very essential because literature evidence suggest that their living conditions expose them to spread of infectious diseases [7,8], which has the potential to impact their psychological wellbeing.

The results, similar to known evidence in the general population, shows that infectious diseases such as COVID-19 impose stressful situations on camped refugees [47–51]. The findings clearly show a high level of anxiety among camped refugees in Ghana during the COVID-19 pandemic. Almost four out of every five (75.4%) reported feeling dizzy, lightheaded, or faint, when they read or listened to news on COVID-19. About half (49.3%) reported having difficulty sleeping, 45.6% were distressed or disturbed. More than 30% reported being unable to carry out essential daily activities because of fear of COVID-19. Feeling of hopelessness and physical reactions, such as sweating or heart palpitation were common occurrences amongst about a third of respondents. Feeling of worry of coming into contact with other refugees, stigma associated with contracting the disease and risk of contracting the disease because of living arrangements within the camps, were also common. Similar findings have been reported among other vulnerable populations including those with insecure livelihoods, physically challenged, the elderly, immigrants, minority groups and displaced populations [52–58].

The results of the multivariate analysis showed that most of the socio-demographic characteristics (camp of residence, country of origin, sex, educational attainment, religious affiliation, engagement in economic activity, head of household, and sex of head of household) had no statistically significant association with severe psychological impact of the pandemic. These results suggest that severe psychological impact of the pandemic was not limited to specific

populations within the camps but was a problem in the general population of refugees in the selected camps. The marital status of the respondent, and age of the household head were the only socio-demographic characteristics that were statistically significantly associated with being impacted severely by the pandemic. The results revealed that respondents who were never married had significantly reduced odds of being severely impacted psychologically by the pandemic when compared to those ever married. Whilst a study of adolescent girls in Cox's Bazar, Bangladesh reported similar findings [59], an online household study in the United States of America reported contrary findings [60]. The social, economic and mental health benefits of marriage are widely discussed in the literature [61–65]. However, marriage as a protective factor of the psychological impact of COVID-19 are mixed.

Respondents from households where the head was aged between 30–39 years had statistically significantly reduced odds of being severely affected psychologically by the pandemic when compared to respondents from households where the head was less than 30 years. Also, respondents from households where the head was aged 40 years or older (40–49 year and 50 + years) had reduced odds of being severely psychologically affected by the pandemic, however, the effects were not statistically significantly different when compared to those from households where the head was less than 30 years. Significant associations between age of household head and psychological impact of COVID-19 has also been reported in other studies [66], however, due to differences in age categorisations, the findings are not comparable.

Without accounting for knowledge of the disease and adherence to the preventive measures, the results showed that males had a statistically significant reduced odds of being severely affected by the pandemic when compared with females. These results have also been reported in similar studies in Pakistan, Spain, Saudi Arabia, Italy and China [67–71]. However, in this study, when knowledge of COVID-19 and adherence to the preventive measures were included in the model, the sex of the respondent became statistically insignificant. This indicates that knowledge of the disease and adherence to the COVID-19 protocols nullifies the sex differentials in the psychological impacts of the pandemic on camped refugees.

The study revealed that number of sources of information on COVID-19, the level of knowledge of the disease and adherence to the preventive measures had a significant association with being severely impacted psychologically by the pandemic. Respondents with two sources of information on COVID-19 had statistically significantly reduced odds of having severe psychological impact when compared to those with one source of information. On the contrary, those with three or more sources had increased odds of having severe psychological impact when compared to those with one source of information, however, the differences were not statistically significant. Studies have reported repeated and higher media exposure to elevate COVID-19 related anxiety and stress [72,73]. Overtime, there has been dramatic changes in information diffusion, with internet technology making unprecedented amount of health information accessible to consumers, nonetheless, there are ambiguities associated with many information sources [74,75]. Thus, relying on fewer trusted information sources on COVID-19 could reduce psychological impact of the pandemics.

The findings show that the higher the level of knowledge and adherence, the higher the odds of having severe psychological impact. Studies have reported that Knowledge of COVID-19 and its related preventive measures are essential to reducing the spread of the disease [76–78], as well as the psychological distress associated with it [79,80]. However, this study has shown that among camped refugees, the higher the level of knowledge and adherence to the preventive measures the higher the likelihood of being severely affected psychologically. Other studies have reported similar findings [81,82]. Also, frequent exposure to information has been reported as a risky mental health factor of COVID-19 [83–85]. These findings have been associated with misinformation, particularly fake news from social media [86–88].

Nonetheless, the findings of this study cannot be attributed to misinformation because, an objective measure was used to assess correct knowledge of the disease. This finding can be placed in the context of Grupe and Nitschke (2013) [89] supposition that knowing about a threat, could disrupt ability to mitigate its negative impacts and thus result in anxiety. Vulnerable populations such as camped refugees are often faced with uncertainties [90], which in addition to the COVID-19 pandemic, could diminish how efficiently and effectively they deal with it. A situation where being knowledgeable about the disease but faced with uncertainties could contribute to increased psychological distress.

With regards to adherence to the COVID-19 preventive measures, studies have shown that anxiety is often associated with higher fear of the disease and adoption of protective health behaviours [91–96]. Thus, it is expected that anxiety results in high adherence to preventive measures, although research evidence remains mixed. Whilst some studies have shown positive relationship between anxiety and adherence to preventive measure, others found no or negative association [97,98]. The findings of this study thus concur with others that found that increased anxiety is associated with positive preventive behaviour.

The results further revealed that pre-existing health conditions such as diabetes, hypertension, heart disease and chronic lung disease and also challenges including overcrowding, water shortages, lack of soap and loss of livelihoods had no statistically significant association with having severe psychologically distress of the COVID-19 pandemic. Contrary to these findings, it has been reported that pre-existing medical conditions are associated with increased risk of poor mental health [99–101]. However, the evidence is mixed with studies in British Columbia, Canada, and Greece, similarly to the findings of this study, reporting no significant association between having a pre-existing chronic health condition and severe levels of anxiety or depression symptoms during the COVID-19 pandemic [102,103]. With regards to challenges, in the general population, it has been shown that overcrowding, water shortages, poverty, loss of livelihood, economic and income uncertainties were associated with psychological distress during the COVID-19 pandemic [104–108]. However, for camped refugees in Ghana, this study has shown no association between challenges they face in the camp and severe psychological impact of the COVID-19 pandemic.

The findings of this study suggest that severe psychological distress of the COVID-19 pandemic among camped refugees is not limited to camped refugees of a particularly background characteristics, but is a problem in the general population of camped refugees. Thus, interventions to improve pandemic related psychological impacts on camped refugees should not be selective but targeted all camped refugees. Also, health education in refugee camps should not be aimed at only enhancing knowledge of the disease and adherence to the preventive measures, but also management of psychological distress, as has been shown in this study, the more knowledgeable and adherent, the more psychologically distressed.

## Conclusions

This study examined the psychological impact of the COVID-19 pandemic on camped refugees and the predictors associated with having a severe psychological impact of the pandemic. Nine COVID-19 Anxiety Scale questions were used to examine the psychological state of refugees in the two camps in Ghana. The results showed high levels of anxiety among camped refugees in Ghana during the pandemic. Respondents socio-demographic characteristics that were associated with severe psychological impact of the pandemic were marital status and age of the head of household. The limited number of socio-demographic characteristics associated with severe psychological impact of the pandemic suggest that, the severe psychological effects was not limited to particular groups but was an effect among the general population. The study

also revealed that high knowledge of the disease and adherence to the preventive measures were positively associated with having a severe psychological impact. Further, the study showed that gender differentials in the psychological effect among camped refugees becomes less important when knowledge of the disease and adherence to the preventive measures are taken into account. Pre-existing health conditions and challenges in the camps were observed not to be associated with psychological impact of the pandemic. The findings of the study suggests that interventions to address psychological impact of the pandemic should target the general population of camped refugees and interventions should not only focus on enhancing knowledge of the disease and adherence to the protocols but also management of the psychological effects.

## Author Contributions

**Conceptualization:** Barbara Sakyi, Fiifi Amoako Johnson.

**Data curation:** Barbara Sakyi.

**Formal analysis:** Fiifi Amoako Johnson.

**Methodology:** Fiifi Amoako Johnson.

**Validation:** Barbara Sakyi, Fiifi Amoako Johnson.

**Visualization:** Barbara Sakyi.

**Writing – original draft:** Barbara Sakyi, Fiifi Amoako Johnson.

**Writing – review & editing:** Barbara Sakyi, Fiifi Amoako Johnson.

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
