## [Decision Letter · Decision Letter 0]

7 Sep 2022

PONE-D-22-20247The psychological impact of the COVID-19 pandemic on camped refugees in GhanPLOS ONE

Dear Dr. Amoako Johnson,

Thank you for submitting your manuscript to PLOS ONE. After careful consideration, we feel that it has merit but does not fully meet PLOS ONE’s publication criteria as it currently stands. Therefore, we invite you to submit a revised version of the manuscript that addresses the points raised during the review process.

We look forward to receiving your revised manuscript.

Kind regards,

Guillermo Salinas-Escudero, PhD. MsC.

Academic Editor

PLOS ONE

Journal Requirements:

4. We note that Figure 1 in your submission contain map/satellite images which may be copyrighted. All PLOS content is published under the Creative Commons Attribution License (CC BY 4.0), which means that the manuscript, images, and Supporting Information files will be freely available online, and any third party is permitted to access, download, copy, distribute, and use these materials in any way, even commercially, with proper attribution. For these reasons, we cannot publish previously copyrighted maps or satellite images created using proprietary data, such as Google software (Google Maps, Street View, and Earth). For more information, see our copyright guidelines: http://journals.plos.org/plosone/s/licenses-and-copyright.

a) You may seek permission from the original copyright holder of Figure 1 to publish the content specifically under the CC BY 4.0 license.  

Additional Editor Comments:

The work was evaluated by two reviewers who agree it is relevant. Notwithstanding, it is suggested to make some modifications to increase the quality and clarity of the work.

Reviewers' comments:

Reviewer's Responses to Questions

**Comments to the Author**

1. Is the manuscript technically sound, and do the data support the conclusions?

Reviewer #1: Yes

Reviewer #2: Yes

2. Has the statistical analysis been performed appropriately and rigorously? 

Reviewer #1: Yes

Reviewer #2: Yes

3. Have the authors made all data underlying the findings in their manuscript fully available?

Reviewer #1: No

Reviewer #2: No

4. Is the manuscript presented in an intelligible fashion and written in standard English?

Reviewer #1: Yes

Reviewer #2: Yes

5. Review Comments to the Author

Reviewer #1: Dear authors,

Thank you for your valuable work and efforts.

I have few comments.

Please, Add study design in obvious title and illustrate it.

Also, Add sample type

Add ethical considerations

Warm regards

Reviewer #2: This is an interesting workThis is exciting and novel work. Studying the impact of the COVID-19 pandemic on a particularly vulnerable population becomes highly relevant. The manuscript is generally well written and methodologically meets sufficient quality criteria to be published. However, I suggest reinforcing some points to improve its quality.

1. It is recommended to analyze the statistical differences between refugee camps.

2. Although the aim of the study focuses on refugee camps, the findings indicate that the characteristics related to the camps do not have the expected statistical relevance. Thus, the results may reflect vulnerable populations, regardless of their area of location. It is therefore suggested to discuss in depth whether the reported characteristics may represent other disadvantaged groups or present more evidence about why these data only represent individuals in refugee camps.

3. Explain in greater detail the findings presented in Table 5, models 1 and 2, regarding the categories that are not significant for the variables (age of the head of the household 40 and over, three or more sources of information on COVID, and marital status. Discuss these findings in the corresponding section.

6. PLOS authors have the option to publish the peer review history of their article (what does this mean?). If published, this will include your full peer review and any attached files.

Reviewer #1: No

Reviewer #2: No

---

## [Author Response · Author response to Decision Letter 0]

5 Oct 2022

Response to Reviewer’s comments

PONE-D-22-20247R1 

A cross-sectional study of the psychological impact of the COVID-19 pandemic on camped refugees in Ghana

PLOS ONE

We are grateful to the Editor and Reviewers for providing us with positive feedback and useful suggestions for ‎the revision. The following notes explain how we addressed each comment in the revised ‎manuscript. Please note C1, C2… refer to comments from the Editor and Reviewers and R1, R2… refer to ‎authors response. The corrections are highlighted in yellow shade. ‎

Journal Requirements 

C1. Please ensure that your manuscript meets PLOS ONE's style requirements, including those for file naming. The PLOS ONE style templates can be found at 

R1. Thank you for the comment. The style has been revised to meet PLOS ONE’s style requirements. 

C2. Please provide additional details regarding participant consent. In the ethics statement in the Methods and online submission information, please ensure that you have specified (1) whether consent was informed and (2) what type you obtained (for instance, written or verbal, and if verbal, how it was documented and witnessed). If your study included minors, state whether you obtained consent from parents or guardians. If the need for consent was waived by the ethics committee, please include this information.

R2. Informed consent, explaining the objectives of study and its relevance and assuring respondents of anonymity and confidentiality, was sort before interviews were conducted. Respondents who could read were made to read the informed consent themselves before signing them. For those who could not read, the informed consent was read to them and where necessary interpreted before signing or thumbprinting. Where respondents were below 18 years of age, informed consent was sort from parents/guardians. This has been indicated in the revised manuscript. Please, see page 5, line 20-23 and page 6, line 1 and 2. We have also added the ethical considerations. Please, see page 6, lines 3-13. 

C3. We note that you have indicated that data from this study are available upon request. PLOS only allows data to be available upon request if there are legal or ethical restrictions on sharing data publicly. For information on unacceptable data access restrictions, please see http://journals.plos.org/plosone/s/data-availability#loc-unacceptable-data-access-restrictions. 

R3. Thank you. The data availability statement has been revised based on your suggestions. Please see 20, lines 17-21.

C4. We note that Figure 1 in your submission contain map/satellite images which may be copyrighted. All PLOS content is published under the Creative Commons Attribution License (CC BY 4.0), which means that the manuscript, images, and Supporting Information files will be freely available online, and any third party is permitted to access, download, copy, distribute, and use these materials in any way, even commercially, with proper attribution. For these reasons, we cannot publish previously copyrighted maps or satellite images created using proprietary data, such as Google software (Google Maps, Street View, and Earth). For more information, see our copyright guidelines: http://journals.plos.org/plosone/s/licenses-and-copyright.

a) You may seek permission from the original copyright holder of Figure 1 to publish the content specifically under the CC BY 4.0 license. 

R4. Thank you for the clarification. We have removed the figure from the revised manuscript.

C5. Please review your reference list to ensure that it is complete and correct. If you have cited papers that have been retracted, please include the rationale for doing so in the manuscript text, or remove these references and replace them with relevant current references. Any changes to the reference list should be mentioned in the rebuttal letter that accompanies your revised manuscript. If you need to cite a retracted article, indicate the article’s retracted status in the References list and also include a citation and full reference for the retraction notice.

R5. We have checked all the references to ensure that they are cited in the manuscript. Additional references included at the revision stage are indicated in this document. Please, see pages 5 and 6 of this document.

Comments from Reviewer 1

Reviewer #1: Dear authors, Thank you for your valuable work and efforts. I have few comments.

C6. Please, Add study design in obvious title and illustrate it.

R6. Thank you for the comment. The study design was cross-sectional. This has been included in the revised title. Please, see page 1, line 1. 

C7. Also, Add sample type

R7. A two-stage sampling approach was used to select respondents. At the first stage, a systematic sampling approach was used to select housing units. At the second stage, where more than one eligible respondent was identified in a housing unit, simple random sampling approach was used to select one respondent. This has been clarified in the revised manuscript. Please, see page 6, line 21, 24-25 and page 7, lines 6-7.

C8. Add ethical considerations

R8. Thank you. We have included the ethical considerations in the revised manuscript. Please, see page 5, lines 20-23 and page 6, line 1-13.

Comments from Reviewer 2

Reviewer #2: This is an interesting work. This is exciting and novel work. Studying the impact of the COVID-19 pandemic on a particularly vulnerable population becomes highly relevant. The manuscript is generally well written and methodologically meets sufficient quality criteria to be published. However, I suggest reinforcing some points to improve its quality.

C8. It is recommended to analyze the statistical differences between refugee camps.

R8. Thank you for the comment. We have analysed the differences between the camps and included it in the revised manuscript. The results showed that the camp effects were not statistically significantly different for both the bivariate (Please, see highlighted section of Table 4 and page 11, line 3) and multivariate analysis (since they were not significant, they were not included in Table 5). The camp of residence is also included as a covariate. See page 8, line 22. These are also included among the non-significant covariates in the Discussions. See page 15, line 20.

C9. Although the aim of the study focuses on refugee camps, the findings indicate that the characteristics related to the camps do not have the expected statistical relevance. Thus, the results may reflect vulnerable populations, regardless of their area of location. It is therefore suggested to discuss in depth whether the reported characteristics may represent other disadvantaged groups or present more evidence about why these data only represent individuals in refugee camps.

R9. Thank you for the comment. We noticed that the statement was not clear enough. Since most of the background characteristics were not statistically significant, we intended to imply that the psychological distress of the COVID-19 is not limited to camped refugees of particular background characteristics, for example, by educational attainment, country of origin, sex, etc…but it is a problem in the general population of camped refugees. It is also important to note that, since the data focused on camped refugees we cannot infer to reflect all vulnerable populations. The statement has been modified in the revised manuscript. Please, see page 19, lines 3-7.

C10. Explain in greater detail the findings presented in Table 5, models 1 and 2, regarding the categories that are not significant for the variables (age of the head of the household 40 and over, three or more sources of information on COVID, and marital status. Discuss these findings in the corresponding section.

R10. Thank you. We have provided detailed discussions of the categories that were not significant for the variables age of the household head, number of sources of information and marital status. Please, see page 16, lines 2 – 17 and page 17, lines 3 – 14. The references below have been included in the revised manuscript to support the discussions. 

58. Baird, S., et al., Intersecting disadvantages for married adolescents: Life after marriage pre-and post-Covid-19 in contexts of displacement. Journal of Adolescent Health, 2022. 70(3): p. S86-S96.

59. Jace, C.E. and C.A. Makridis, Does marriage protect mental health? Evidence from the COVID‐19 pandemic. Social Science Quarterly, 2021. 102(6): p. 2499-2515.

60. Ahituv, A. and R.I. Lerman, How do marital status, work effort, and wage rates interact? Demography, 2007. 44(3): p. 623-647.

61. Glenn, N.D. and C.N. Weaver, The contribution of marital happiness to global happiness. Journal of Marriage and the Family, 1981: p. 161-168.

62. Gove, W.R., M. Hughes, and C.B. Style, Does marriage have positive effects on the psychological well-being of the individual? Journal of health and social behavior, 1983: p. 122-131.

63. Kiecolt-Glaser, J.K. and T.L. Newton, Marriage and health: his and hers. Psychological bulletin, 2001. 127(4): p. 472.

64. Lillard, L.A. and L.J. Waite, 'Til death do us part: Marital disruption and mortality. American journal of sociology, 1995. 100(5): p. 1131-1156.

65. Islam, M.M. and M. Alharthi, Impact of COVID-19 on the Quality of Life of Households in Saudi Arabia. International Journal of Environmental Research and Public Health, 2022. 19(3): p. 1538.

71. Alrasheed, M., S. Alrasheed, and A.S. Alqahtani, Impact of Social Media Exposure on Risk Perceptions, Mental Health Outcomes, and Preventive Behaviors during the COVID-19 Pandemic in Saudi Arabia. Saudi Journal of Health Systems Research: p. 1-7.

72. Garfin, D.R., R.C. Silver, and E.A. Holman, The novel coronavirus (COVID-2019) outbreak: Amplification of public health consequences by media exposure. Health psychology, 2020. 39(5): p. 355.

73. Han, R., J. Xu, and D. Pan, How Media Exposure, Media Trust, and Media Bias Perception Influence Public Evaluation of COVID-19 Pandemic in International Metropolises. International Journal of Environmental Research and Public Health, 2022. 19(7): p. 3942.

74. Purvis, R.S., et al., Perceptions of adult Arkansans regarding trusted sources of information about the COVID-19 pandemic. BMC Public Health, 2021. 21(1): p. 1-9.

---

## [Decision Letter · Decision Letter 1]

18 Oct 2022

PONE-D-22-20247R1A cross-sectional study of the psychological impact of the COVID-19 pandemic on camped refugees in GhanaPLOS ONE

Dear Dr. Amoako Johnson,

Thank you for submitting your manuscript to PLOS ONE. After careful consideration, we feel that it has merit but does not fully meet PLOS ONE’s publication criteria as it currently stands. Therefore, we invite you to submit a revised version of the manuscript that addresses the points raised during the review process.

We look forward to receiving your revised manuscript.

Kind regards,

Guillermo Salinas-Escudero, PhD. MsC.

Academic Editor

PLOS ONE

Journal Requirements:

Additional Editor Comments:

Add the comparison between both groups for each variable presented in table four. Ensure to add a column with the correspondent significance level.

Reviewers' comments:

Reviewer's Responses to Questions

**Comments to the Author**

1. If the authors have adequately addressed your comments raised in a previous round of review and you feel that this manuscript is now acceptable for publication, you may indicate that here to bypass the “Comments to the Author” section, enter your conflict of interest statement in the “Confidential to Editor” section, and submit your "Accept" recommendation.

Reviewer #1: (No Response)

Reviewer #2: All comments have been addressed

2. Is the manuscript technically sound, and do the data support the conclusions?

Reviewer #1: Yes

Reviewer #2: Yes

3. Has the statistical analysis been performed appropriately and rigorously? 

Reviewer #1: Yes

Reviewer #2: Yes

4. Have the authors made all data underlying the findings in their manuscript fully available?

Reviewer #1: No

Reviewer #2: Yes

5. Is the manuscript presented in an intelligible fashion and written in standard English?

Reviewer #1: Yes

Reviewer #2: Yes

6. Review Comments to the Author

Reviewer #1: (No Response)

Reviewer #2: Please note that the statistical differences between refugee camps need to be done for each variable.

7. PLOS authors have the option to publish the peer review history of their article (what does this mean?). If published, this will include your full peer review and any attached files.

Reviewer #1: No

Reviewer #2: No

---

## [Author Response · Author response to Decision Letter 1]

20 Oct 2022

Response to Reviewer’s comments

PONE-D-22-20247R1 

A cross-sectional study of the psychological impact of the COVID-19 pandemic on camped refugees in Ghana

PLOS ONE

We are once again grateful to the Editor and Reviewers for their comments and suggestions. The following notes explain how we addressed the comments in the revised ‎manuscript. Please note C1, C2… refer to comments from the Editor and Reviewers and R1, R2… refer to ‎authors response. The corrections are highlighted in yellow shade. ‎

Journal Requirements 

C1. Please review your reference list to ensure that it is complete and correct. If you have cited papers that have been retracted, please include the rationale for doing so in the manuscript text, or remove these references and replace them with relevant current references. Any changes to the reference list should be mentioned in the rebuttal letter that accompanies your revised manuscript. If you need to cite a retracted article, indicate the article’s retracted status in the References list and also include a citation and full reference for the retraction notice

R1. Thank you, We have checked the reference list and confirm that they are complete and correct. To address C2, the reference below was added to support the discussion. Please, see page 9, lines 16 – 17 and page 26, lines 4 – 5. 

46. Kim, H.-Y., Statistical notes for clinical researchers: Sample size calculation 2. Comparison of two independent proportions. Restorative Dentistry & Endodontics, 2016. 41(2): p. 154-156.

Additional Editor Comments 

C2. Add the comparison between both groups for each variable presented in table four. Ensure to add a column with the correspondent significance level.

Thank you for the suggestion. The suggested analysis has been included in the revised manuscript. Please see pages 35 and 36, highlighted section of Table 4. Please, note that Fisher's Exact Test was used to test for significant differences where cell sample size were small (less than five). The reference below has been included to support the decision. Please, see page 9, lines 16 – 17. The results are discussed in the highlighted sections of pages 11 – 13. 

46. Kim, H.-Y., Statistical notes for clinical researchers: Sample size calculation 2. Comparison of two independent proportions. Restorative Dentistry & Endodontics, 2016. 41(2): p. 154-156.

---

## [Editor Report · Decision Letter 2]

25 Oct 2022

PONE-D-22-20247R2A cross-sectional study of the psychological impact of the COVID-19 pandemic on camped refugees in GhanaPLOS ONE

Dear Dr. Amoako Johnson,

Thank you for submitting your manuscript to PLOS ONE. After careful consideration, we feel that it has merit but does not fully meet PLOS ONE’s publication criteria as it currently stands. Therefore, we invite you to submit a revised version of the manuscript that addresses the points raised during the review process.

We look forward to receiving your revised manuscript.

Kind regards,

Guillermo Salinas-Escudero, PhD. MsC.

Academic Editor

PLOS ONE

Journal Requirements:

Additional Editor Comments (if provided):

Reviewer 2 was requesting the comparison of each variable of table 4 between the 2 refugees camps, to evaluate whether the values between camps are different or not (significance level), unfortunately, an analysis of the differences between the different categories of the variable for all the fields, which does not correspond to the reviewer's request.

Therefore, it is forwarded for correction,
---

## [Author Response · Author response to Decision Letter 2]

27 Oct 2022

Response to Reviewer’s comments

PONE-D-22-20247R3 

A cross-sectional study of the psychological impact of the COVID-19 pandemic on camped refugees in Ghana

PLOS ONE

We are grateful to the Editor and Reviewers for their comments and suggestions. The following notes explain how we addressed the comments in the revised ‎manuscript. Please note C1, C2… refer to comments from the Editor and Reviewers and R1, R2… refer to ‎authors response. The corrections are highlighted in yellow shade. ‎

Journal Requirements 

C1. Please review your reference list to ensure that it is complete and correct. If you have cited papers that have been retracted, please include the rationale for doing so in the manuscript text, or remove these references and replace them with relevant current references. Any changes to the reference list should be mentioned in the rebuttal letter that accompanies your revised manuscript. If you need to cite a retracted article, indicate the article’s retracted status in the References list and also include a citation and full reference for the retraction notice.

R1. Thank you, we have checked the reference list and confirm that they are complete and correct. 

Additional Editor Comments 

C2. Reviewer 2 was requesting the comparison of each variable of table 4 between the 2 refugees camps, to evaluate whether the values between camps are different or not (significance level), unfortunately, an analysis of the differences between the different categories of the variable for all the fields, which does not correspond to the reviewer's request. Therefore, it is forwarded for correction.

R2. Thank you for the clarity. We have conducted both within and between camp analysis for the two refugees camps. We have included the p-values which examines the significance of the different categories of each variable for within and between the two camps in the revised manuscript. Please, see highlighted section of Table 4 and highlighted sections of page 11 – 15 . This is also clarified in the statistical analysis section. Please, see page 9, line 16 – 20.

---

## [Decision Letter · Decision Letter 3]

31 Oct 2022

A cross-sectional study of the psychological impact of the COVID-19 pandemic on camped refugees in Ghana

PONE-D-22-20247R3

Dear Dr. Amoako Johnson,

We’re pleased to inform you that your manuscript has been judged scientifically suitable for publication and will be formally accepted for publication once it meets all outstanding technical requirements.

Kind regards,

Guillermo Salinas-Escudero, PhD. MsC.

Academic Editor

PLOS ONE

Additional Editor Comments (optional):

The work was reviewed and approved by both reviewers. However, it is recommended to include reviewer two's recommendations to improve the quality of the manuscript.

Reviewers' comments:

Reviewer's Responses to Questions

**Comments to the Author**

1. If the authors have adequately addressed your comments raised in a previous round of review and you feel that this manuscript is now acceptable for publication, you may indicate that here to bypass the “Comments to the Author” section, enter your conflict of interest statement in the “Confidential to Editor” section, and submit your "Accept" recommendation.

Reviewer #2: All comments have been addressed

2. Is the manuscript technically sound, and do the data support the conclusions?

Reviewer #2: Yes

3. Has the statistical analysis been performed appropriately and rigorously? 

Reviewer #2: Yes

4. Have the authors made all data underlying the findings in their manuscript fully available?

Reviewer #2: Yes

5. Is the manuscript presented in an intelligible fashion and written in standard English?

Reviewer #2: Yes

6. Review Comments to the Author

Reviewer #2: Thanks for responding to my comment. It seems, however, that I did not express my request very well. It does not make sense to present a test of differences between each variable category. What is relevant is to know if there are significant differences for each variable between groups. Thus, I suggest analyzing differences for each variable between groups. Also, please make it clear in the table's foot which test of difference of means or proportions was performed.

7. PLOS authors have the option to publish the peer review history of their article (what does this mean?). If published, this will include your full peer review and any attached files.

Reviewer #2: No

---

## [Editor Report · Acceptance letter]

4 Nov 2022

PONE-D-22-20247R3 

A cross-sectional study of the psychological impact of the COVID-19 pandemic on camped refugees in Ghana 

Dear Dr. Amoako Johnson:

I'm pleased to inform you that your manuscript has been deemed suitable for publication in PLOS ONE. Congratulations! Your manuscript is now with our production department. 

Kind regards, 

on behalf of

Dr. Guillermo Salinas-Escudero 

Academic Editor

PLOS ONE